# Computational Investigation of the pH Dependence of Stability of Melanosome Proteins: Implication for Melanosome formation and Disease

**DOI:** 10.3390/ijms22158273

**Published:** 2021-07-31

**Authors:** Mahesh Koirala, H. B. Mihiri Shashikala, Jacob Jeffries, Bohua Wu, Stacie K. Loftus, Jonathan H. Zippin, Emil Alexov

**Affiliations:** 1Department of Physics, Clemson University, Clemson, SC 29634, USA; mkoiral@g.clemson.edu (M.K.); mhewabo@g.clemson.edu (H.B.M.S.); jwjeffr@g.clemson.edu (J.J.); bohua@g.clemson.edu (B.W.); 2Genetic Disease Research Branch, National Human Genome Research Branch, National Institutes of Health, Bethesda, MD 22066, USA; sloftus@mail.nih.gov; 3Department of Dermatology, Weill Cornell Medical College, New York, NY 10021, USA; jhzippin@med.cornell.edu

**Keywords:** pH dependence, proton transport, pH regulation, stability

## Abstract

Intravesicular pH plays a crucial role in melanosome maturation and function. Melanosomal pH changes during maturation from very acidic in the early stages to neutral in late stages. Neutral pH is critical for providing optimal conditions for the rate-limiting, pH-sensitive melanin-synthesizing enzyme tyrosinase (TYR). This dramatic change in pH is thought to result from the activity of several proteins that control melanosomal pH. Here, we computationally investigated the pH-dependent stability of several melanosomal membrane proteins and compared them to the pH dependence of the stability of TYR. We confirmed that the pH optimum of TYR is neutral, and we also found that proteins that are negative regulators of melanosomal pH are predicted to function optimally at neutral pH. In contrast, positive pH regulators were predicted to have an acidic pH optimum. We propose a competitive mechanism among positive and negative regulators that results in pH equilibrium. Our findings are consistent with previous work that demonstrated a correlation between the pH optima of stability and activity, and they are consistent with the expected activity of positive and negative regulators of melanosomal pH. Furthermore, our data suggest that disease-causing variants impact the pH dependence of melanosomal proteins; this is particularly prominent for the OCA2 protein. In conclusion, melanosomal pH appears to affect the activity of multiple melanosomal proteins.

## 1. Introduction

The pH of a solution is an important characteristic for many biological processes. On a molecular level, pH controls macromolecular stability and, at extreme pH (acidic or basic extremes), macromolecules unfold. Typically, for every macromolecule, there is a particular pH at which the macromolecule is the most stable and activity is maximum, termed the pH optimum [1,2]. Macromolecular interactions are also pH-dependent [3,4,5], and there is typically a pH optimum at which the binding affinity is maximum [4]. Within a cell, subcellular compartments have different pH, reflecting their function, from low pH in lysosomes to high pH in peroxisomes. Thus, macromolecules tend to have a pH optimum that is ideal for the pH of the subcellular compartment where they reside [3]. Increasing the scale of this idea, pH plays a crucial role for body organ function and varies from very acidic in the stomach to neutral in the blood. All above examples indicate that the regulation and maintenance of pH is essential for many biological phenomena. 

pH is maintained in a given cellular compartment by channels and/or pumps either by directly trafficking H^+^ or by indirectly affecting the local H^+^ concentration. These channels and/or pumps can be termed positive (increase pH) or negative (decrease pH) regulators [6,7]. Reaching and maintaining the desired pH depends on the balance of H^+^ flux controlled by these regulators, including passive transport across the membrane (Figure 1). One would expect that the positive regulators are most active at acidic pH and show almost no activity at basic pH since their role is to increase pH. The converse would be expected for negative regulators, whereby activity increases as the pH rises. At a particular pH, the inward and outward flux of H^+^ ions induced by positive and negative regulators becomes equal and the pH setpoint is established (Figure 1).

Melanocytes are a specialized cell type that resides in the skin, eyes, brain, ears, heart, lungs, and adipose tissue [8]. One of the primary functions of melanocytes is the production of melanin, a polymer of tyrosine derivatives that has important chemical properties in a wide range of tissues [9]. Melanin is synthesized in a specialized organelle called the melanosome. The pH of this organelle varies during the development of the organelle (a multistage process called maturation) and contributes to common pigmentation variation in human skin, hair, and eye color. Biallelic rare variants in proteins critical for the production of melanin (e.g., TYR) or in pH regulation of the melanosome (e.g., OCA2 and SLC45A2) lead to a significant reduction in melanin pigmentation in the skin, eyes, and hair and give rise to oculocutaneous albinism (OCA) (OCA1, OCA2, and OCA4, respectively). Melanin synthesis is critical for the protection of the skin and eyes from ultraviolet radiation; a reduction in melanin synthesis increases the risk of skin cancers. Furthermore, a dramatic reduction in melanin production in the eye is also associated with foveal hypoplasia, reduced visual acuity, and photophobia among individuals with OCA [10]. Taken together, the link between altered melanin pigment production and disease is well documented; however, it remains poorly understood how the pH of this organelle affects function of proteins critical for the maintenance of organelle pH [11].

Melanosomes originate from the endosome (Figure 2); thus, early melanosomes have a low pH (~3–4), whereas, during maturation, the pH reaches a near-neutral pH of about 7. The near-neutral pH of the mature melanosome is thought to provide a favorable environment for tyrosinase (TYR), the rate-limiting melanin-synthesizing enzyme [12,13,14]. The change in pH during melanosome maturation is thought to be controlled by several membrane proteins [7] (e.g., OCA2, SLC45A2, and TPC2/TPCN2) (Figure 2). OCA2 and SLC45A2 are presumed to be positive pH regulators, while TPC2 is considered to be a negative pH regulator. Considering the proposed role of positive and negative pH regulators (Figure 1), we anticipated that these proteins have different pH profiles of stability and activity. Whereas we highlight these proteins because of their association with pigment disease and published studies, it should be mentioned that other melanosome proteins may also be pH-sensitive but there are currently no genetic or functional data to support our computational examination. In addition, there are other melanosome proteins important for melanin synthesis (e.g., the ATP7A protein, which is altered in individuals with Menkes disease, and functions to supply Cu^2+^ to the melanosome for TYR catalytic activity) that may exhibit pH-dependent stability and activity. We hypothesized that the ATP7A protein would have a similar pH dependence to TYR [15,16].

We anticipated that OCA2 and SLC45A2 would have maximal activity at acidic pH, whereas TPC2 would have maximal activity at basic pH. OCA2 plays a major role in eye color variation [17] and regulates melanosomal pH and maturation [17,18,19]. It may also be involved in small-molecule transport for the biosynthesis of melanin [20,21]. SLC45A2 also participates in the transport of substances required for melanin biosynthesis [19,22,23]. TPC2 affects pigmentation by regulating melanosome pH and size by mediating Ca^2+^ release from the organelle [24,25].

Thus, understanding how melanosomal pH affects the activity of these proteins is essential. Furthermore, these proteins are commonly mutated in disease, and those variants may impact the normal pH optimum of these proteins. Predicting the pH optimum of activity is not an easy task and requires modeling of the details of the corresponding biochemical reactions as a function of pH. Here, we take advantage of the observation that the pH optima of activity and stability are typically the same, as indicated in our earlier work [12]. Our goal was to computationally determine the pH dependence of stability of OCA2, SLC45A2, TPC2, TYR, and ATP7A proteins, as well as the effects of genetic variant alleles on their pH-dependent stability.

## 2. Results

As pointed out earlier, in this work, we focus on several proteins participating in melanosome formation, with the goal of contrasting their stability pH dependence and the effect of pathogenic variants. Using methods established in our previous work on the correlation between pH optimum of activity and the pH optimum of stability [1], we now predict protein stability over varying pH and infer the activity of each protein. We present the results according to the classification of the proteins as “positive” and “negative” regulators and probe our hypothesis that “positive” regulators should have a lower pH optimum as compared with “negative” regulators. Firstly, we present the results of the wild-type proteins, followed by the results of the mutants. 

### 2.1. pH Dependence of Folding Free Energy on Wild-Type Proteins

For each wild-type protein, the magnitude of the “constant” in Equation (1) is unknown, because there are no experimental data of the folding free energy (the difference of the free energy of folded and unfolded states) [26,27,28] at a given pH for any of the proteins modeled in this work. Accordingly, it was set to zero at the beginning of the simulated pH interval, pH = 4.0. Here, we present the calculated pH dependence of the folding free energy using an energy-minimized structure (Figure 3), and we averaged results over 20 snapshots taken from MD simulations (Appendix A). We do not focus heavily on the results obtained with MD snapshots because DelPhiPKa was developed to calculate the pKa of ionizable groups using static structures. However, we probe the sensitivity of the results using MD snapshots to investigate the role of plausible conformational changes on the pH dependence of stability. We saw no significant differences in the results obtained with the energy-minimized structure and the averaged results over 20 snapshots, suggesting that there are no structural changes contributing to the stability pH dependence. 

TYR had the highest pH optimum of stability, about 8.0 or higher (Figure 3). In contrast, the OCA2 protein (predicted to be a positive regulator of pH) had the lowest pH optimum of stability, about 5.0–6.0. The other predicted positive regulator, the SLC45A2 protein, also had a pH optimum lower that neutral pH, i.e., 6.5. The presumed negative regulator TPC2 and the ATP7A protein which supplies copper to TYR both had pH optima close to neutral pH. Thus, there was a distinctive predicted difference in pH stability of OCA2, SLC45A2, TYR, ATP7A, and TPC2. Furthermore, our modeling confirmed the experimental observations that TYR is most active at neutral pH with reduced activity at acidic pH [14].

The pH dependence of the folding free energy is derived from the difference in pKa of ionizable groups in the folded and unfolded state. Thus, if the pKa in folded and unfolded states is the same, there would be no pH dependence. Furthermore, if the pKa is only different when outside the pH region of interest, the pH dependence of the folding free energy would still be affected but may not be physiologically relevant. It is not expected that the pKa of titratable groups in the unfolded state would be perturbed from standard pKa values [29]; thus, most of the pH dependence of the folding free energy should originate from a perturbed pKa in the folded state. However, for completeness, in Appendix A, we provide the calculated pKa for both states, folded and unfolded. Indeed, one can see that, for “positive” regulators, most of perturbed pKa occurrences are for acidic groups, thus resulting in pH dependence at low pH. In contrast, most of perturbed pKa occurrences for TYR, ATP7, and the “negative” regulator TCP2 are of His residues, resulting in pH dependence at neutral pH. 

### 2.2. Effect of Pathogenic Variants on Protein Stability

Table 1 shows the average change in folding free energy due to variants according to predictions made using the methods described above. The low standard deviations reported reflect the consistency of results obtained with different tools. Most of the variants were predicted to destabilize the proteins by a modest amount. However, some variants, such as A481T and N489D in OCA2, as well as C1002F and I1264V in ATP7A, were predicted to significantly affect protein stability. In the case of the OCA2 A481T and N489D variants, both of which have been observed among individuals with albinism, the predicted large change in folding free energy can be attributed to the change in the physicochemical properties of the wild-type residues: A→T and N→D. A→T represents a hydrophobic to polar residue change, while N→D represents a polar to charged residue change. In contrast, C1002F and I1264V variants in ATP7A are conservative but were also predicted to result in a large change in folding energy. In this case, the change in folding energy is thought to be caused by the distortion of the residue packing caused by the different geometries of the side-chains [30]. The structures of the proteins with variant sites mapped are provided in Appendix A.

Overall, the predicted changes in the folding free energy were not extremely large; however, since we do not know the absolute folding free energy of the proteins and how the change in protein stability affects the activity, it is impossible to definitively know how these moderate changes affect protein activity. However, we can reasonably assume that protein activity will decrease when folding free energy changes, even when the variants appear to make the protein more stable (e.g., H615L in OCA2 protein), because, in most cases, any significant deviation of wild-type properties is deleterious for protein function [31,32].

### 2.3. pH Dependence of Folding Free Energy on Genetic Variation

We predicted the effect of nonsynonymous variants on the pH dependence of protein stability by comparing the wild-type and corresponding variant proteins using both free energy-minimized structures (Figure 4) and snapshots generated via MD simulations (Appendix A). One can see that there was no significant difference in the results obtained with different protocols. As mentioned in Section 4, we considered that the “constant” in Equation (1) was the predicted folding free energy change caused by the variants (Table 1). The most drastic effects were found for OCA2, whereas variants in other proteins had moderate effects on the pH dependence of folding free energy. In the case of OCA2, most of the variants (except one, H615L) were predicted to alter the pH dependence of stability, suggesting that the variant proteins would be less stable at neutral pH.

Furthermore, many variants in OCA2 (R419Q, N489D, and V443I) resulted in a shift in pH optimum to lower pH. This would result in a shift in maximal activity of OCA2 toward the lower pH range and could result in a shift in the balance between positive and negative regulators such that the resulting pH setpoint would be lower than the wild-type melanosome. Lower pH in the melanosome would result in reduced TYR activity, which is found in patients with OCA2.

The above observations focused on the shape of the pH dependence curve of folding free energy without considering the magnitude of the change. It should be mentioned that the changes in the folding free energy of OCA2 were within several kcal/mol, while the changes in the pH dependence of folding free energy caused by variants in other proteins were sometimes larger (Figure 4). Despite this, the predicted changes in protein stability would likely affect protein activity and alter melanosome pH.

The reasons why variants in OCA2 had significant effects on the pH dependence of folding free energy can be found in Appendix A. Our study focused on the pH interval 4.0 to 8.0, and the pH dependence was predicted to be due to titratable groups over this pH range that have different pKa values in the folded versus unfolded state. Such titratable groups are Asp, Glu, and His. One can see that, in the case of OCA2, variants resulted in a perturbed pKa of Glu and Asp, while having almost no effect on the pKa of His. This is the reason why the pH dependence of the folding free energy of OCA2 was mostly affected over acidic pH.

## 3. Discussions

We studied wild-type and genetic variants of melanosome and melanin-synthesizing proteins [19] to predict the pH dependence of their folding free energy. Consistent with previous reports, TYR was determined to be very pH-sensitive. In TYR, neither of the two variants, R402Q and S192Y, affected pH dependence, either because they did not involve titratable groups or involved titratable groups with very high pKa, or because their pH dependence was outside the pH interval of the study. Of note, a single haplotype allele in which these two alleles, R402Q and S192Y, are found together in *cis* is linked to a pathogenic haplotype for OCA1 [33]. Therefore, we examined whether the presence of both variants affected protein stability; TYR modeled with both variants had a modest change in protein stability (Table 1). In the case of ATP7A, modest changes in the pH dependence caused by variants were predicted to occur at neutral and higher pH. Overall, the changes in stability were quite small. The reason for modest changes in the stability with little effect on the pH dependence could be attributed to the conservative nature of the variants. In all cases, the physicochemical properties of the wild-type sites were preserved. Considering SLC45A2, neither variant involved titratable groups or caused alteration of the wild-type pH dependence of the folding free energy. However, significant changes in protein stability were predicted, which would affect SLC45A2 function. Significant alterations in the pH dependence of the folding free energy caused by variants were predicted for the OCA2 protein. Indeed, most of the variants altered the wild-type physicochemical properties of the protein. As a result, all variants had a shift in pH optimum from the wild-type OCA2. Lastly, the variants in TPC2 did not cause significant changes in the stability or pH dependence of the folding free energy.

Our data suggest that variants predicted to be pathogenic (e.g., OCA2: N489D, V443I) likely function by affecting protein stability and/or the pH dependence of folding free energy; our data also identified known variants with conflicting interpretations/unknown significance (e.g., OCA2: A481T, R419Q) that may affect protein stability and/or the pH dependence of folding free energy. *OCA2**R419Q is thought to modify the penetrance of the OCA2 locus and may affect the risk of melanoma [34]; therefore, our data may support a role for melanosomal pH in melanoma genesis. Our predictions regarding these variants require biological testing to confirm our conclusions. It should be noted that our analysis failed to predict any pH or protein stability effects of other variants predicted to be pathogenic (e.g., TYR: R402Q, OCA2: P743L, or ATP7A: G666R). Thus, our data do not support the foregone conclusion that variants in protein-coding sequences affect protein stability and/or pH dependence of folding free energy; thus, other mechanisms of protein inactivation should be explored. The TPC2 variants (K376R, M484I, and V219I) are conservative variants; thus, their identification by GWAS suggests that these variants may reside in LD with other variants or structural alleles that impact expression, protein stability, or protein function. Furthermore, SLC45A2 (rs16891982 = L374F) was a top SNV associated with altered SLC45A2 mRNA expression levels and may mediate the GWAS linkage association via this mechanism [35]. Given the uncertainty of variant associations with function, our data suggest that assessing the function of protein variants on a large scale using structural modeling may be helpful. Perhaps, the addition of a pH polygenic score that takes into consideration the pH impact on the melanosome and all of its channels and enzymes will help in the assignment of variants to predicted functional groups.

## 4. Materials and Methods

This section consists of four components: (1) obtaining 3D structures of the proteins of interest, (2) generation of mutants in silico, (3) molecular dynamics simulations, and (4) calculating the pH dependence of the folding free energy. 

### 4.1. Structures Used in the Modeling

TYR protein: The 3D structure of TYR was modeled using SWISS-MODEL [36] from an amino-acid sequence of length (529 aa) taken from UniProt (ID: P14679) [37]. A template (PDB ID: 5M8P) [38] with an identity of 44%, covering 81% (19–452) of the total sequence of TYR was selected. The corresponding model and template are shown in Appendix A. 

OCA2 protein: The 3D structure of OCA2 was modeled using Phyre2 [39]. The full-length sequence of OCA2 is 838 aa and was taken from UniProt (ID: Q04671) [37]. A template (PDB ID: 4F35) [40] was selected with an identity of 20% to query, covering 60% of the sequence of OCA2 (Appendix A). The helical content was well preserved between the template and the model (Appendix A).

TPC2 protein: A crystal structure for TPC2 is available (PDB ID: 6NQ2) [41] and is a homodimer with 752 residues (Appendix A).

SLC45A2 protein: The 3D structure of SLC45A2 was modeled using Phyre2 [39]. Its sequence was taken from UniProt (ID: Q9UMX9) [37] with a sequence length of 530 amino acids. The chosen template (PDB ID: 4YBQ) [42] covered 94% of the sequence with an identity of 14% (Appendix A). 

ATP7A protein: The 3D modeling of this protein was also done using Phyre2 [39]. The sequence was taken from UniProt (ID: Q04656) [37] with a sequence length of 1500 amino acids. The template (PDB ID: 3RFU) [43] covered 57% of the sequence (646–1411) with an identity of 41% (Appendix A).

### 4.2. List of Nonsynonymous GWAS-Identified Pigmentation-Associated Variants

The NHGRI-EBI catalog of human genome-wide association studies (GWAS) [44] was queried on 4 April 2020 to identify all nonsynonymous variants in genes TYR, OCA2, SLC45A2, TPCN2, and ATP7A found associated with common human pigmentation variation of skin and hair (see Table 1).

Of note, the variants identified by GWAS were associations. In the case of nonsynonymous coding variants, they may impact protein function or, alternatively, may, similarly to those associations identified in noncoding regions of the genome, be in tight linkage disequilibrium (LD) with other variants that may function to impact expression levels or proper splicing. These studies are important as they measure the impact of protein variation on the pH dependence of the folding free energy and can help to establish whether a variant directly impacts the protein in question.

### 4.3. Generation of Mutants

To generate the 3D structure of protein variants while avoiding the introduction of artificial errors, we used the model of the wild-type protein, and the corresponding residue was mutated using UCSF Chimera [45]. The folded wild-type structures and variant sites mapped onto a 3D structure of folded TYR, OCA2, TPC2, and ATP7A are shown in Appendix A. One can see (Appendix A) that most of the variants were within well-preserved structural regions, away from the loops, which reduced the uncertainty of the 3D modeling.

### 4.4. Molecular Dynamics (MD) Simulations 

MD simulations were performed under periodic boundary conditions using NAMD2.9 [46] with atomic parameters of the CHARMM force field [47]. The protein structures were prepared for the simulations using VMD [48], and TIP3P water molecules were applied to build the explicit water solvated systems. Finally, the system was neutralized with NaCl when necessary.

Simulations were performed for 20 ns for each protein structure with different initial atomic velocities. In the production stages of the simulations, they were equilibrated under constant volume–temperature (NVT) conditions for 100 ps followed by 2000 ps (2 ns) of constant pressure–temperature (NPT) equilibration at 1 atm pressure and 310 K (with the same restraints). The first 15 ns of the simulations were not equilibrated; thus, they were removed. The structural analysis was sampled from the last 5 ns at every 250 ps. This produced 20 snapshots per structure, all of which were subjected to DelPhiPKa [49,50,51] calculations after removing the explicit water molecules.

### 4.5. Modeling pH Dependence of Folding Free Energy

To model the pH dependence of folding free energy, we built a 3D model of the unfolded state [29]. The unfolded structure ensembles of the wild-type proteins were generated using the “flexible meccano” approach [52,53], and, among them, we selected one representative structure (the structure with no helices and strands) (Appendix A). The unfolded mutants were then generated using UCSF Chimera [45].

The pKa and net charge of the wild-type protein and mutants, in both the folded and the unfolded states, were calculated using DelPhiPKa [49,50,51]. We also calculated the pKa and net charge for each of the 20 snapshots taken from the MD simulation to obtain the average net charge and its difference with respect to wild-type proteins.

The change in folding free energy (ΔΔG ^folding^) was calculated from the net charge difference between the folded state and the unfolded state, taking the unfolded state as the initial state. The following equation was used over the pH range of interest, giving an explicit pH-dependent form of the folding free energy [1,54]:(1)$$\Delta \Delta G^{folding}=2.3kT\sum \Delta qdpH+constant,$$
where Δq is the change in net charge from the unfolded to folded state, and dpH is the pH interval. The constant is the absolute folding free energy at a given pH.

For the analysis of the wild-type proteins, the “constant” was considered to be zero at the beginning of the pH interval because there is no information about the absolute folding free energy of the individual proteins and predicting it would introduce significant and unwanted noise. However, for the mutants, there are many algorithms that are benchmarked against experimental data and shown to perform well, which gives us the opportunity to predict the change in folding free energy caused by a variant with acceptable confidence. Thus, for the mutants, the “constant” was considered to be the free energy difference between wild-type and mutant proteins caused by the variant. The folding free energy changes were modeled using an in-house algorithm, the SAAFEC-SEQ [28] method, along with third-party tools such as INPS3D [55], INPS-SEQ [55], mCSM [56], SDM [57], DUET [58], I-Mutant-SEQ [59], MUpro-SEQ [60], iStable-SEQ [61], and DeepDDG [62].

## 5. Conclusions

The importance of melanosomal pH for the regulation of organelle maturation is well studied [20,24]; however, how melanosomal proteins or their genetic variants [24] regulate pH remains unknown. Here, we proposed a mechanism of the competitive pH dependence of stability and activity of “positive” and “negative” pH regulators. Our data suggest that the predicted “positive” regulators of melanosomal pH have maximal activity (and, thus, maximal stability) at low pH, while the opposite is predicted from “negative” regulators. Our data suggest that OCA2 and SLC45A2 have low pH optima as compared to the TPC2 protein. Furthermore, TYR and ATP7A are predicted to have pH optima at neutral and/or higher pH. We speculate that similar mechanisms of pH regulation are expected for other melanosomal proteins.

## Figures and Tables

**Figure 1 ijms-22-08273-f001:**
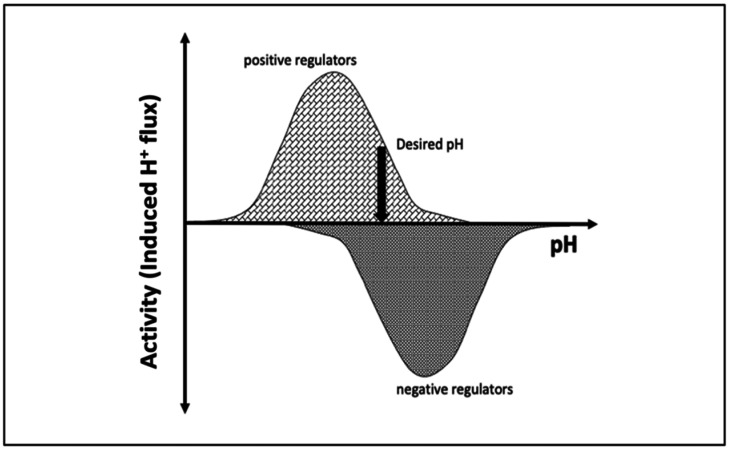
Schematic representation of the induced H^+^ flux of positive (increase pH) and negative (decrease pH) regulators. The vertical arrow indicates the desired pH, at which the total induced H^+^ flux is zero.

**Figure 2 ijms-22-08273-f002:**
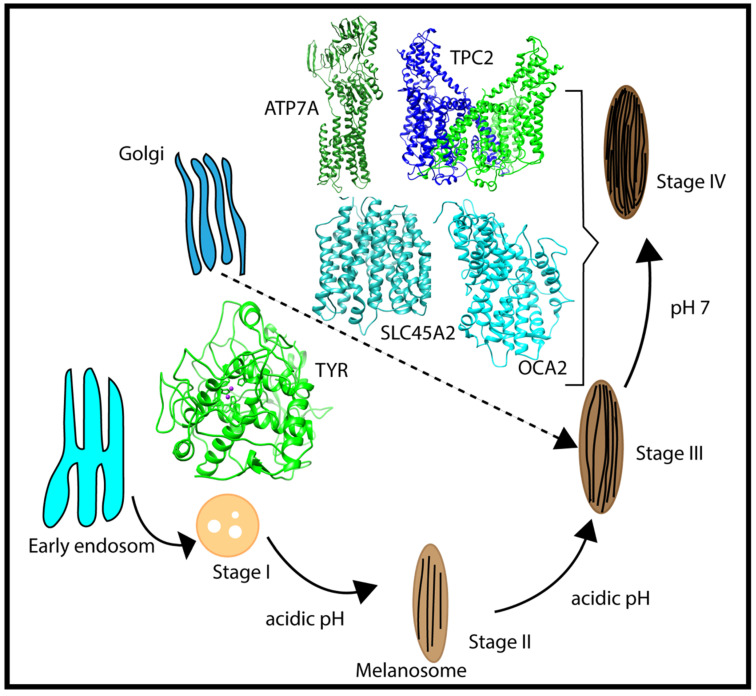
Schematic representation of the multistage processes of melanosome formation and proteins participating in pH regulation and melanin synthesis. The characteristic pH for each melanosome stage is also indicated in the figure.

**Figure 3 ijms-22-08273-f003:**
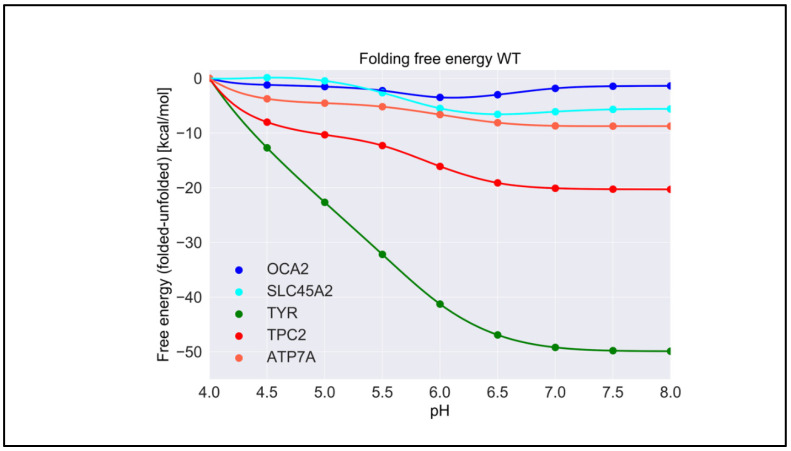
The pH dependence of the folding free energy of wild-type proteins from minimized structures within pH range 4–8.

**Figure 4 ijms-22-08273-f004:**
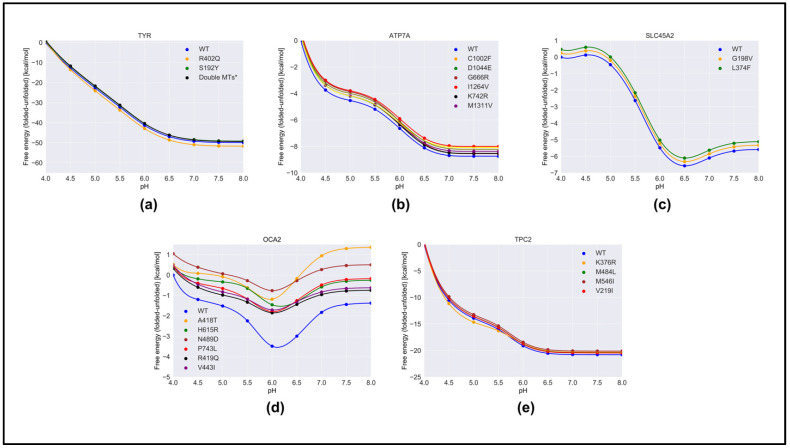
The pH dependence of the folding free energy of wild-type proteins and their mutants from minimized structures within pH range 4–8. (**a**)TYR; (**b**) ATP7A; (**c**) SLC45A2; (**d**) OCA2 & (**e**) TPC2.

**Table 1 ijms-22-08273-t001:** Change in folding free energy due to variants.

Change in Folding Free Energy (ΔΔG) Due to Variants(kcal/mol)			
Protein	Variant	Avg ƊΔG	SD
TYR	R402Q	−0.5	0.5
	S192 YDouble MT *	−0.27 −0.77	0.78 1.09
OCA2	A481T	−1.01	0.52
	H615L	0.17	0.39
	N489D	−1.05	1.08
	P743L	−0.9	0.45
	R419Q	−0.54	0.33
	V443I	−0.54	0.48
SLC45A2	G198V	−0.51	0.25
	L374F	−0.84	0.47
TPC2	K376R	−0.49	0.3
	M484L	−0.86	0.33
	M546I	−0.1	0.67
	V219I	−0.11	0.32
ATP7A	C1002F	−1.2	0.74
	G666R	−0.21	0.7
	D1044E	−0.8	0.53
	I1264V	−1.1	0.74
	K742R	0.01	0.35
	M1311V	−0.79	0.35
	R844C	−0.48	0.39
	S653Y	−0.45	0.54

Note: Positive and negative values of ΔΔG represent stabilization and destabilization due to the variant, respectively. The asterisk indicates a double mutant (R402Q and S192Y) for TYR, where ΔΔG was calculated by taking the sum of individual changes.

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
