# Peer review of "Computational Investigation of the pH Dependence of Stability of Melanosome Proteins: Implication for Melanosome formation and Disease"

_ijms, 2021, doi:10.3390/ijms22158273_

Round 1
Reviewer 1 Report
Melanosomal pH is critical for understanding the process of melanosomal maturation. The authors of this manuscript are trying to answer this question using molecular modeling. They suggest that pH optimum in melanosomes is a balance of positive and negative pH regulators. In this work, melanosomal proteins tyrosinase, OCA2, SLC45A2, ATP7A, and TPCN2 were suggested as potential pH regulators.
I have several concerns:
- More than 1500 proteins function in the melanosome. For me, it is not clear why these five proteins authors selected from the melanosomal protein crowd. In addition, the pH level in melanosomes is affected by melanins and other products of the reaction. Why these five proteins is a leading force in pH regulation?
- In this work, four protein models were obtained using the Phyre2 server. Each of these models was incomplete structurally. How it is possible to calculate the correct pH levels from incomplete structures?
- It is not a well-documented study. Most of the details of the quality of homology modeling, computer simulations, and protein PDB files were not presented.
- The absence of experimental data.
Author Response
We thank the reviewer for their useful comments and suggestions. Below we address reviewer’s questions point by point:
Rev: More than 1500 proteins function in the melanosome. For me, it is not clear why these five proteins authors selected from the melanosomal protein crowd.
Answer: Sorry for not clarifying this point in the manuscript. The proteins studied were chosen because they are either known to directly affect melanosome pH (e.g., OCA2) or are known to be sensitive to changes in melanosome pH levels (e.g., TYR). We would expect that other melanosome proteins are also pH sensitive but at this time we do not have the structural or functional data to support their examination. We have now clarified this point further in the revision – lines 65-66.
Rev: In addition, the pH level in melanosomes is affected by melanins and other products of the reaction. Why these five proteins is a leading force in pH regulation?
Answer: We are not aware of any role for melanin itself in the regulation of melanosome pH. There are reports that DOPA, a product of melanin synthesis, is a ligand for the melanosome protein GPR143. However, the function GPR143 is not well established. We do not mean to suggest that these proteins are the leading force in pH regulation. However, there is a large body of functional evidence that OCA2, SLC45A2 and TPCN2 control melanosome pH; therefore, it is reasonable to focus our analysis on these three proteins. TYR was included because it is a well-established pH sensitive melanosome protein but it does not regulate pH itself. TYR, in a sense, is used as a positive control for our modeling. ATP7A was included because it has a critical role in melanin synthesis, little is known about how pH might affect the activity of this protein, and the structural information was available for evaluation.
Rev: In this work, four protein models were obtained using the Phyre2 server. Each of these models was incomplete structurally. How it is possible to calculate the correct pH levels from incomplete structures?
Answer: The models were built on highly similar templates (as demonstrated in the supplementary material) and thus they are expected to be of high quality. Furthermore, the software that was used to calculate pH-dependence, the DelPhiPKa, operates using Gaussian-based presentation of the protein properties, which reduces the effect of plausible structural imperfections. Lastly, the results obtained with the 3D model of TYR protein match available experimental data, thus assuring that pH-dependence calculations are adequate even using protein models.
Rev: It is not a well-documented study. Most of the details of the quality of homology modeling, computer simulations, and protein PDB files were not presented.
Answer: All details are provided in the Supplementary material. These include sequence alignment, structural superimposition and all other relevant details.
Rev: The absence of experimental data.
Answer: This is completely computational study. Our data makes predictions regarding the impact of pH on protein function. Some of our conclusions are substantiated by published functional data (e.g., TYR). We do plan to confirm our computational predictions but that is beyond the scope of this current study.
English was improved as well.
Reviewer 2 Report
This article presents a theoretical study of melanosome proteins, but it is not clear from both the title and the abstract of the article that this is a completely theoretical work. The main remark for the paper is that the authors use the term folding (free) energy, but there is no such concept in protein physics. The authors of the article should correct this in the text and all figures. Until this is done, it is impossible to understand the results of this work. Authors should indicate errors of methods and programs used in the work.
The authors use the TPC2 protein in the text, and TPCN2 in the figure.
Author Response
We thank the reviewer for their useful comments and suggestions. Below we address reviewer’s questions point by point:
Rev: This article presents a theoretical study of melanosome proteins, but it is not clear from both the title and the abstract of the article that this is a completely theoretical work.
Answer: Thank you for this helpful comment. We have corrected both the title and the abstract (line 14).
Rev: The main remark for the paper is that the authors use the term folding (free) energy, but there is no such concept in protein physics.
Answer: The term “folding free energy” is widely used in molecular biophysics to refer to the free energy difference between the unfolded and folded states of macromolecules (proteins). However, in light of the reviewer’s comment, we have provided a definition and several references at the first instance we use the term “folding free energy” to ensure that the reader is educated.
Rev: The authors of the article should correct this in the text and all figures. Until this is done, it is impossible to understand the results of this work. Authors should indicate errors of methods and programs used in the work.
Answer: We have revised the manuscript to address this critique. Regarding errors, the DelPhiPKa method, which was used to compute pH-dependence, is completely deterministic. Thus, given the structures of proteins, it calculates the same pH-dependence and does not result in variations. Therefore, statistical evaluation of the errors cannot be carried out. All programs used in the work are clearly mentioned in the revision.
Rev: The authors use the TPC2 protein in the text, and TPCN2 in the figure.
Answer: We have corrected this inconsistency.
English was improved as well.
Round 2
Reviewer 1 Report
accept in present form
Reviewer 2 Report
I have no comments.